# Provision of Psychological Support to a 31-Year-Old Man with SARS-CoV2-Induced Pneumonia during and after Hospitalization: A Clinical Case Report

**DOI:** 10.3390/ijerph20010757

**Published:** 2022-12-31

**Authors:** Edward Callus

**Affiliations:** 1Clinical Psychology Service, IRCCS Policlinico San Donato, San Donato Milanese, 20097 Milan, Italy; 2Department of Biomedical Sciences for Health, University of Milan, 20133 Milan, Italy; edward.callus@unimi.it; Tel.: +39-025-277-4645

**Keywords:** case report, pandemic, COVID-19, depression, anxiety, post-traumatic stress disorder, insomnia, clinical psychology, remote interventions

## Abstract

A 31-year-old man hospitalized during the first wave of the pandemic in 2020 suffering from severe psychological distress, requested psychological assistance as his condition progressively worsened, eventually requiring intubation. After being referred to the clinical psychology service by a ward physician, the patient was assisted remotely for two months for a total of 22 sessions during hospitalization and after discharge. A psychometric evaluation was carried out when the patient was close to discharge and longitudinally, for a total of four times, for depression (Patient Health Questionnaire-9 (PHQ-9)), anxiety (Generalized Anxiety Disorder Scale-7 (GAD-7)), post-traumatic stress disorder (Impact of Event Scale—Revised (IES-R)) and insomnia (Insomnia Severity Index (ISI)). Support was provided remotely, mainly through audio and video calls, and text chats were also utilized when possible and as required. The initial psychometric evaluation indicated moderate depression, severe anxiety, the presence of post-traumatic stress disorder and sleep problems. Psychological distress decreased until reaching a situation of no distress in the final evaluation. Psychological interventions from which the patient benefitted were stress reduction and breathing techniques, empathic support, elaboration of the possibility of grief and cognitive restructuring regarding fears relative to his condition. Psychological support provided remotely and the monitoring of psychological status after discharge are highly advisable in pandemic emergency situations. The CARE checklist of information to include when writing a case report was utilized in the writeup of this case report.

## 1. Introduction

A 31-year-old man was admitted to the hospital’s emergency room during the peak of the pandemic in early March 2020 in Milan following an episode of loss of consciousness resulting in a head injury. In the three days prior to accessing the emergency room, the patient reported fever, even though during hospitalization, his temperature was normal, and his breathing was regular, with an absence of coughing or wheezing. The next day, the positive diagnosis of COVID-19 was confirmed through a nasopharyngeal swab.

In the following days, the patient’s condition worsened following the onset of SARS-CoV2-induced pneumonia, which resulted in excessive overregulation of cytokine production (cytokine storm). Initially, a respiratory assistance procedure was implemented through continuous positive airway pressure (CPAP) with subsequent induction of a coma state preparatory to intubation.

The patient explicitly requested the possibility of psychological support to a physician in the ward. He was experiencing high levels of anxiety and panic due to breathing difficulties, insomnia and fear of death experienced during CPAP care. He had previously sought psychotherapeutic care, mainly to improve his intimate relationships, and he reported having obtained benefit from it, concluding treatment in December 2019. No major psychological and/or psychiatric disturbances were reported prior to his hospitalization.

The experience of hospitalization following COVID-19 infection is frequently associated with the appearance of significant levels of anxiety [1,2], depression, sleep disorders [3,4] and symptomatology compatible with the onset of post-traumatic stress disorder [5].

The case described herein therefore represents an example of treatment with the aim of assisting an emotionally and physically fragile patient, whose expectations related to hope of survival and recovery were negatively affected by the severity of his health condition.

Initially, relaxation and guided breathing techniques [6] were utilized to diminish psychological distress. Subsequently, the patient was supported remotely during the following months, up to hospital discharge and at home.

In an extensive search of the relevant databases, no similar description of a similar case study was found in the literature. Such reports could be of help for clinical psychologists who find themselves having to assist patients remotely in similar critical situations.

## 2. Materials and Methods

The first contact with the patient occurred 3 days after his hospitalization. All sessions were organized remotely through the use of phone calls or video calls through WhatsApp, as well as through WhatsApp chats when the patient did not have the necessary strength to speak, following all guidelines and recommendations, indicating that clinical psychology services should be organized online and/or via telephone [7] because non-essential personnel, such as psychiatrists, psychologists and social workers, are strongly discouraged from entering isolation wards in these situations.

An Excel database was utilized to report the days and times of each session, and notes were taken in a Word document after each session. These documents, together with the record of the text messages on WhatsApp (which were removed from the author’s mobile device for privacy reasons [8]) are stored on computers protected by password to ensure patient confidentiality. 

Psychometric testing was utilized to assess and monitor the patient’s psychological functioning and as an indicator of the effectiveness of the psychological intervention as soon his physical and psychological condition permitted him complete the questionnaire. The questionnaires were administered by sending the patient a link which led to online forms which could be filled in directly from his device. 

The only face-to face session was the last one, and it occurred 66 days after the patient’s discharge from hospital.

### 2.1. Diagnostic Assessment

In accordance with the literature cited in the Introduction, the following tests were selected for psychometric assessment of both hospitalized and discharged COVID-19 patients: **a)** The Generalized Anxiety Disorder Scale-7 (GAD-7) [9], a screening tool for generalized anxiety disorders in clinical practice and research. In addition, it provides a measure of severity and is linked to the criteria of the Diagnostic and Statistical Manual of Mental Disorders (DSM-IV);**b)** The Patient Health Questionnaire-9 (PHQ-9) [10], a short psychological screening tool designed to measure symptoms of depression in primary care facilities;**c)** The Impact of Event Scale - Revised (IES-R) [11], a self-report measure (DSM-IV) that assesses subjective distress caused by traumatic events;**d)** The Insomnia Severity Index (ISI) [12], a self-report questionnaire that assesses the nature, severity and impact of insomnia.

The patient was assisted remotely for two months for a total of 22 sessions during hospitalization and after discharge for a total 12 h and 29 min (excluding the last face-to-face session). The first psychometric assessment was carried out approximately 1 month and 5 days after the patient’s hospitalization, just before he was discharged from the hospital.

### 2.2. Therapeutic Intervention

During the critical phase, the focus was on the management of the very high anxiety, which was amplified by difficulties in breathing. In the first session, the patient was already using CPAP, and this caused anxiety and panic due to the discomfort created by the strong jet of the air blowing and the noise of the machine. 

The patient also requested the support of psychotropic drugs to manage anxiety levels, which were, unfortunately, refused due to his clinical condition. The only alternative that remained was psychoeducation through breathing and relaxation exercises. 

Considering the critical situation and the physical condition of the patient, which did not enable concentration for long stretches of time, as well as high anxiety levels, it was agreed to establish contact 3 times per day. The sessions were brief and focused on symptom management and provision of support. The patient showed considerable improvement in terms of anxiety symptom management after the initial sessions. 

As his medical condition worsened, physicians anticipated the necessity of intubation to manage the situation. In this phase, the interventions helped the patient to articulate fears and questions regarding his condition and to facilitate communication with the healthcare staff, who were considerably burdened due to the sudden pandemic onset. 

When the necessity of intubation became more probable, the focus of the interventions was to support the patient on how to handle communication with his partner and his family. High emotional distress, physical fragility, and physical isolation from the people he cared for acted as very important barriers and created a sense of disorientation regarding what to say and to whom in this extremely critical moment. 

It was therefore suggested to work on a written document that I could deliver to his loved ones in the case of the worst outcomes. At the same time, the sessions were focalized on increasing emotional awareness and supporting the patient in his decisions on what to communicate and to whom during this critical time. 

The patient was subsequently intubated for 12 days. During the post-intubation phase, the patient reported disorientation and difficulty in reconstructing what had happened to him. In this phase, we agreed on one session daily. The patient reported feeling emotionally overwhelmed and having a lot of fear about the possibility of long-term outcomes resulting from being infected. The first psychometric assessment was carried out at this point. 

During this period, the focus of the sessions was on the management of post-hospitalization complications, in particular the management of anxiety levels and catastrophic thoughts. In some specialized medical visits, there was a contemplation of the utilization of psychotropic drugs (a physician suggested this during a follow-up visit), mainly for the management of anxiety; however, the patient decided to proceed only with psychological support. After two weeks and 4 days, the second assessment was carried out, and there was an indication that the psychological situation was improving. It was therefore decided to reduce the sessions to once every 2 days. 

The main theme of the next sessions was the patient’s return to work and his previous commitments. There was a gradual shift from the management of post-traumatic aspects to handling issues concerning everyday life, in particular, to establish healthy boundaries and the capacity to say no to some requests. 

The sessions were gradually reduced to twice a week, then once a week. In this phase, after about three weeks, the third assessment took place, which confirmed the absence of post-traumatic stress disorder (PTSD) symptoms and improvement in psychological functioning. 

The sessions were concluded with a face-to-face session (the only one), in which it was established that the support could be concluded, as also confirmed by the final evaluation carried out about two weeks after the previous one, which indicated that no symptoms of distress were present.

## 3. Results

As indicated in the table below (Table 1), during the first assessment (after approximately one month of hospitalization), the patient reported having moderate depression, severe anxiety, a possible presence of PTSD and some problems linked to insomnia, which were below the clinical threshold. After 2 weeks and 4 days, the levels of anxiety and depression decreased considerably, reaching a mild level. The levels of PTSD and insomnia problems remained the same. About three weeks later there was a slight improvement in mood and insomnia levels, and the patient reported being free from PTSD and mood symptoms. In the final assessment, after about 2 weeks, no psychological problems were reported. 

## 4. Discussion

The experience of hospitalization resulting from COVID-19 infection is frequently associated with the appearance of significant levels of psychological distress in response to the dynamics of isolation within the hospital environment and the methods of treatment of associated respiratory problems [3,4]. The presence of a helmet for respiratory assistance in this case led to strong constant noises and difficulties in finding a position in bed favorable to falling asleep for the patient, a condition responsible for the amplification of emotions such as fear and a sense of loneliness, among others. In cases in which the treatment of the pathology requires invasive procedures (i.e., CPAP), it is common to observe the expression of symptoms of PTSD [5].

When caring for such critical patients, integrating psychological and personal variables in medical databases, for example, with patient-reported outcomes, can potentially improve the power of eventual decision support systems. The patients might also feel better cared-for if their perspective is included, possibly having a positive influence on their emotional wellbeing and improving communication between the physician and the patient [13].

Anxiety, depressive symptoms and impaired sleep behavior are the main expressions of psychological distress generally reported by patients [1,2], as confirmed by the results obtained from the initial psychometric evaluation of the patient. Based on the discomfort expressed by the patient and the request for psychological intervention, the choice of a non-pharmacological treatment focused on the use of relaxation techniques and breathing exercises was proven effective in reducing anxiety levels, as previously confirmed in the literature [14].

It is not surprising that patients who face these kinds of situations experience high emotional distress [15]. In this specific case, the previous utilization of psychotherapeutic support may have helped the patient to formulate the request for psychological assistance and to use it with confidence and benefit. This would not be the case for many other patients, who could be both less literate when it comes to psychological interventions and feel stigma or shame when it comes to psychological support.

Another important aspect to consider is that in these very critical periods, healthcare staff is extremely burdened and lack time; therefore, referral of patients to psychological support is probably not optimal, especially in cases in which education is required. For this reason, during the progression of the pandemic and the further waves, we organized to contact all patients who were hospitalized due to COVID-19, presenting ourselves (by calling the patients on their phone, or the ward phone/tablet when not available) and explaining what kind of support we could provide.

One important difference in the handling of this case compared to other hospitalized patients was the change in the setting, in terms of both the temporal and spatial dimensions and the modalities with which the psychological support was provided. Fortunately, the patient did not have any pre-existent psychiatric conditions, which could have complicated the handling of the absence of stronger temporal and physical boundaries. The critical situation demanded an important flexibility in this sense, and after the critical phase, the patient was gradually supported to reach a psychological equilibrium, at which point the support was no longer needed.

Apart from the adequate referral, it is important to ensure that the necessary human resources to provide psychological support are present in medical departments when these situations occur and that they are implemented under conditions of safety and with the possibility of technically providing the required services. In this particular case, the entire clinical psychology service team was heavily exposed to the virus; therefore, I found myself in a situation in which I was supporting a patient who was suffering tremendously from a virus to which I was exposed to. This could have potentially interfered with the efficacy of the psychological support that was provided. Daily team supervisions and support between colleagues were paramount to create the necessary protective factors to enable the staff to function well.

### Patient Perspective

The patient was highly motivated to receive psychological support and reported benefit from the intervention, both through the patient-reported outcomes and by expressing his gratitude during the sessions. In the follow-up phase, the patient reported greater awareness about his emotions and a better management of anxiety stress levels by progressively learning to apply breathing and relaxation exercises independently.

## 5. Conclusions

It is highly advisable to provide remote structured and systematic psychological support provided for patients who are hospitalized during a pandemic. The psychological distress that often accompanies the experience of dealing with acute health situations is compounded by isolation and difficulties in breathing, which potentially amplify levels of anxiety and panic. Psychological support is even more essential, as psychotropic drugs are often not advisable in patients with such low saturation levels. 

When providing care in critical conditions, the patients’ psychological wellbeing should be carefully monitored with the inclusion of patient-reported outcomes administered longitudinally when the patients’ conditions allow for this. 

Support should be offered to all patients during the hospitalization phase and during follow-up after discharge, with quantitative and qualitative monitoring of eventual psychological distress levels.

Finally, mental health professionals taking care of patients in acute settings also need to take care of their own mental health with frequent checks with colleagues and/or supervisors. 

## Figures and Tables

**Table 1 ijerph-20-00757-t001:** Cutoffs and results of the longitudinal psychometric assessment.

Test	Cutoff	27/3/2020	14/4/2020	07/5/2020	22/5/2020
PHQ ^1^	0–4, absent 5–9, below threshold 10–14, mild 15–19, moderate ≥ 20, severe Cutoff, ≥5	Moderate	Mild	Absent	Absent
GAD-7 ^2^	0–4, normal 5–9, mild 10–14, moderate ≥15, severe Cutoff ≥5	Severe	Mild	Mild	In the norm
IES-R ^3^	Low scores indicate better operation. A total score of 33 or higher out of a maximum score of 88 indicates the probable presence of PTSD.	Probable presence of PTSD	Probable presence of PTSD	No PTSD	No PTSD
ISS ^4^	0–7 = absence of clinically significant insomnia 8–14 = insomnia below the clinical threshold 15–21 = clinical insomnia (medium severity) 22–28 = clinical (severe) insomnia Cutoff, ≥8	Problems below the clinical threshold	Problems below the clinical threshold	No sleep problems	No sleep problems

Note: ^1^ Generalized Anxiety Disorder Scale-7; ^2^ Patient Health Questionnaire–9; ^3^ Impact of Event Scale; ^4^ Insomnia Severity Index.

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
