# Peer review of "Provision of Psychological Support to a 31-Year-Old Man with SARS-CoV2-Induced Pneumonia during and after Hospitalization: A Clinical Case Report"

_ijerph, 2022, doi:10.3390/ijerph20010757_

Round 1

Reviewer 1 Report

The paper presents an interesting and timely case study about providing psychological and emotional support to a patient during an acute phase of illness.  It presents important lessons learned in response to delivering remote psychological support during the Covid-19 pandemic at a time when clinicians had to adapt quickly in response to infection.  More information about how confidentiality was ensure, for example, when using Whatsapp with the patient would be useful.  It would be interesting for the author to embed this work within the broader growing evidence base for providing a  better psychologically informed approach to medical care and broader lessons learned. 

Author Response

Dear reviewer, many thanks for your constructive feedback which I believe helped me to improve the case report. I added information about confidentiality as requested and I added a reference specifically on WhatsApp data. The documents which were used to write the case study are always accessed from computers which are protected by passwords. WhatsApp live interactions are considered secure, and the text messages were downloaded and are stored on the password protected PCs and they were eliminated from the author’s device. A relevant article regarding WhatsApp data was cited (Lines 69-73).

I included an additional paragraph on how to proceed to enable psychologically informed approach to medical care (lines 175-180) including a pertinent reference on the personalisation of healthcare. 

Broader lessons learned were included by expanding the conclusions (Lines 232-240). 

Reviewer 2 Report

The mental health of patients with COVID-19 is today very important issue and we still need studies considering medical and psychological support. An author of this case report describes psychological support to a 31-year-old man with SARS-COV2-induced pneumonia during and after hospitalization. Patient was using CPAP and subsequently intubated during hospitalization. Psychological help was held remotely for patient due to epidemiological rules. Also, psychometric evaluation was done.

The paper is well written and organized.

The paper covers an interesting topic and can be helpful for healthcare providers.

The recommendation would be giving the information (material and methods section) about the time of first session (on which day after admission to the hospital) and the time of last session (on which day after discharge from the hospital).

Author Response

Dear reviewer, thank you for your feedback. All the information you requested has been added to the manuscript  (lines 62, 78 and 79). Many thanks as I feel that this has improved the understanding of the timeline of events.